# Cross-Domain Transferability of Adversarial Perturbations

**Muzammal Naseer**[1,2], **Salman Khan**[2,1], **Muhammad Haris Khan**[2],
**Fahad Shahbaz Khan**[2,3], **Fatih Porikli**[1]
[1]Australian National University, Canberra, Australia
[2]Inception Institute of Artificial Intelligence, Abu Dhabi, UAE
[3]CVL, Department of Electrical Engineering, Linköping University, Sweden
`{muzammal.naseer,fatih.porikli}@anu.edu.au`
`{salman.khan,muhammad.haris,fahad.khan}@inceptioniai.org`

## Abstract

Adversarial examples reveal the blind spots of deep neural networks (DNNs) and represent a major concern for security-critical applications. The transferability of adversarial examples makes real-world attacks possible in black-box settings, where the attacker is forbidden to access the internal parameters of the model. The underlying assumption in most adversary generation methods, whether learning an instance-specific or an instance-agnostic perturbation, is the direct or indirect reliance on the original domain-specific data distribution. In this work, for the first time, we demonstrate the existence of domain-invariant adversaries, thereby showing common adversarial space among different datasets and models. To this end, we propose a framework capable of launching highly transferable attacks that crafts adversarial patterns to mislead networks trained on entirely different domains. For instance, an adversarial function learned on Paintings, Cartoons or Medical images can successfully perturb ImageNet samples to fool the classifier, with success rates as high as ~99% ($\ell_\infty \leq 10$). The core of our proposed adversarial function is a generative network that is trained using a relativistic supervisory signal that enables domain-invariant perturbations. Our approach sets the new state-of-the-art for fooling rates, both under the white-box and black-box scenarios. Furthermore, despite being an instance-agnostic perturbation function, our attack outperforms the conventionally much stronger instance-specific attack methods. Code is available at: `https://github.com/Muzammal-Naseer/Cross-domain-perturbations`

## 1 Introduction

Albeit displaying remarkable performance across a range of tasks, Deep Neural Networks (DNNs) are highly vulnerable to adversarial examples, which are carefully crafted examples generated by adding a certain degree of noise (a.k.a. perturbations) to the corresponding original images, typically appearing quasi-imperceptible to humans [1]. Importantly, these adversarial examples are transferable from one network to another, even when the other network fashions a different architecture and possibly trained on a different subset of training data [2, 3]. Transferability permits an adversarial attack, without knowing the internals of the target network, posing serious security concerns on the practical deployment of these models.

Adversarial perturbations are either *instance-specific* or *instance-agnostic*. The instance-specific attacks iteratively optimize a perturbation pattern specific to an input sample (e.g., [4, 5, 6, 7, 8, 9, 10, 11]). In comparison, the instance-agnostic attacks learn a universal perturbation or a function that finds adversarial patterns on a data distribution instead of a single sample. For example, [12]

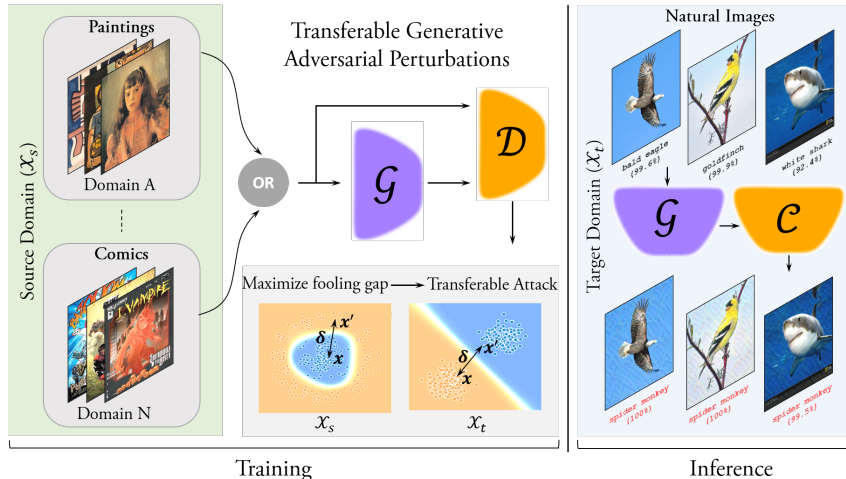

Figure 1: *Transferable Generative Adversarial Perturbation:* We demonstrate that common adversaries exist across different image domains and introduce a highly transferable attack approach that carefully crafts adversarial patterns to fool classifiers trained on totally different domains. Our generative scheme learns to reconstruct adversaries on paintings or comics (*left*) that can successfully fool natural image classifiers with high fooling rates at the inference time (*right*).

proposed universal adversarial perturbations that can fool a model on the majority of the source dataset images. To reduce dependency on the input data samples, [13] maximizes layer activations of the source network while [14] extracts deluding perturbations using class impressions relying on the source label space. To enhance the transferability of instance-agnostic approaches, recent generative models attempt to directly craft perturbations using an adversarially trained function [15, 16].

We observe that most prior works on crafting adversarial attacks suffer from two pivotal limitations that restrict their transferability to real-world scenarios. **(a)** Existing attacks rely directly or indirectly on the source (training) data, which hampers their transferability to other domains. From a practical standpoint, source domain can be unknown, or the domain-specific data may be unavailable to the attacker. Therefore, a true "*black-box*" attack must be able to fool learned models across different target domains without ever being explicitly trained on those data domains. **(b)** Instance-agnostic attacks, compared with their counterparts, are far more scalable to large datasets as they avoid expensive per-instance iterative optimization. However, they demonstrate weaker transferability rates than the instance-specific attacks. Consequently, the design of highly transferable instance-agnostic attacks that also generalize across unseen domains is largely an unsolved problem.

In this work, we introduce '*domain-agnostic*' generation of adversarial examples, with the aim of relaxing the source data reliance assumption. In particular, we propose a flexible framework capable of launching vastly transferable adversarial attacks, e.g., perturbations found on paintings, comics or medical images are shown to trick natural image classifiers trained on ImageNet dataset with high fooling rates. A distinguishing feature of our approach is the introduction of relativistic loss that explicitly enforces learning of domain-invariant adversarial patterns. Our attack algorithm is highly scalable to large-scale datasets since it learns a universal adversarial function that avoids expensive iterative optimization from instance-specific attacks. While enjoying the efficient inference time of instance-agnostic methods, our algorithm outperforms all existing attack methods (both instance-specific and agnostic) by a significant margin ($\sim 86.46\%$ average increase in fooling rate from naturally trained Inception-v3 to adversarially trained models in comparison to state-of-the-art [10]) and sets the new state-of-the-art under both white-box and black-box settings. Figure 1 provides an overview of our approach.

## 2   Related Work

**Image-dependent Perturbations:** Several approaches target creation of image-dependent perturbations. [17] noticed that despite exhibiting impressive performance, neural networks can be fooled through maliciously crafted perturbations that appear quasi-imperceptible to humans. Following this

finding, many approaches [4, 5, 6, 7, 8, 9] investigate the existence of these perturbations. They either apply gradient ascent in the pixel space or solve complex optimizations. Recently, a few methods [18, 10] propose input or gradient transformation modules to improve the transferability of adversarial examples. A common characteristic of the aforementioned approaches is their data-dependence; the perturbations are computed for each data-point separately in a mutually exclusive way. Further, these approaches render inefficiently at inference time since they iterate on the input multiple times. In contrast, we resort to a data-independent approach based on a generator, demonstrating improved inference-time efficiency along with high transferability rates.

**Universal Adversarial Perturbation:** Seminal work of [12] introduces the existence of Universal Adversarial Perturbation (UAP). It is a single noise vector which when added to a data-point can fool a pretrained model. [12] crafts UAP in an iterative fashion utilizing target data-points that is capable of flipping their labels. Though it can generate image-agnostic UAP, the success ratio of their attack is proportional to the number of training samples used for crafting UAP. [13] proposes a so-called data-independent algorithm by maximizing the product of mean activations at multiple layers given a universal perturbation as input. This method crafts a so-called data-independent perturbation, however, the attack success ratio is not comparable to [12]. Instead, we propose a fully distribution-agnostic approach that crafts adversarial examples directly from a learned generator, as opposed to first generating perturbations followed by their addition to images.

**Generator-oriented Perturbations:** Another branch of attacks leverage generative models to craft adversaries. [15] learns a generator network to perturb images, however, the unbounded perturbation magnitude in their case might render perceptible perturbations at test time. [19] trains conditional generators to learn original data manifold and searches the latent space conditioned on the human recognizable target class that is mis-classified by a target classier. [20] apply generative adversarial networks to craft visually realistic perturbations and build distilled network to perform black-box attack. Similarly, [16, 14] train generators to create adversaries to launch attacks; the former uses target data directly and the latter relies on class impressions.

A common trait of prior work is that they either rely directly (or indirectly) upon the data distribution and/or entail access to its label space for creating adversarial examples (Table 1). In contrast, we propose a flexible, distribution-agnostic approach - inculcating relativistic loss - to craft adversarial examples that achieves state-of-the-art results both under white-box and black-box attack settings.

| Method | Data Type | Transfer Strength | Label Agnostic | Cross-domain Attack |
|--------|-----------|-------------------|----------------|---------------------|
| FFF [13] | Pretrained-net/data | Low | ✓ | ✗ |
| AAA [14] | Class Impressions | Medium | ✗ | ✗ |
| UAP [12] | ImageNet | Low | ✗ | ✗ |
| GAP [16] | ImageNet | Medium | ✗ | ✗ |
| RHP [11] | ImageNet | Medium | ✗ | ✗ |
| Ours | Arbitrary (Paintings, Comics, Medical scans etc.) | High | ✓ | ✓ |

Table 1: A comparison of different attack methods based on their dependency on data distribution and labels.

## 3 Cross-Domain Transferable Perturbations

Our proposed approach is based on a generative model that is trained using an adversarial mechanism. Assume we have an input image $x_s$ belonging to a source domain $\mathcal{X}_s \in \mathbb{R}^s$. We aim to train a universal function that learns to add a perturbation pattern $\delta$ on the source domain which can successfully fool a network trained on source $\mathcal{X}_s$ as well as any target domain $\mathcal{X}_t \subset \mathbb{R}^t$ when fed with perturbed inputs $x'_t = x_t + \delta$. Importantly, our training is only performed on the unlabelled source domain dataset with $n_s$ samples: $\{x_s^i\}_{i=1}^{n_s}$ and the target domain is not used at all during training. For brevity, in the following discussion, we will only refer the input and perturbed images using $x$ and $x'$ respectively and the domain will be clear from the context.

The proposed framework consists of a generator $\mathcal{G}_\theta(x)$ and a discriminator $\mathcal{D}_\psi(x)$ parameterized by $\theta$ and $\psi$. In our case, we initialize discriminator with a pretrained network and the parameters $\psi$ are remained fixed while the $\mathcal{G}_\theta$ is learned. The output of $\mathcal{G}_\theta$ is scaled to have a fixed norm and it lies within a bound; $x' = \text{clip}\big(\min(x + \epsilon, \max(\mathcal{G}_\theta(x), x - \epsilon))\big)$. The perturbed images $x'$ as well as

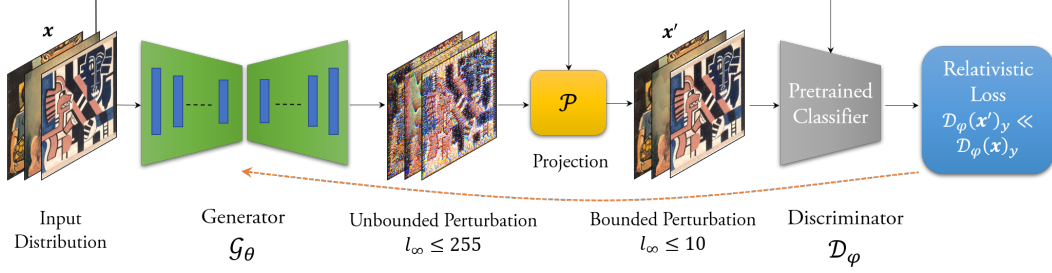

| Input Distribution | Generator $\mathcal{G}_\theta$ | Unbounded Perturbation $l_\infty \leq 255$ | Bounded Perturbation $l_\infty \leq 10$ | Discriminator $\mathcal{D}_\varphi$ |

Figure 2: The proposed generative framework seeks to maximize the '*fooling gap*' that helps in achieving very high transferability rates across domains. The orange dashed line shows the flow of gradients, notably only the generator is tuned in the whole pipeline to fool the pretrained discriminator.

the real images $\boldsymbol{x}$ are passed through the discriminator. The output of the discriminator denotes the class probabilities $\mathcal{D}_\psi(\boldsymbol{x}, \boldsymbol{x}') \in [0, 1]^c$, where $c$ is the number of classes. This is different from the traditional GAN framework where a discriminator only estimate whether an input is real or fake. For an adversarial attack, the goal is to fool a network on most examples by making minor changes to its inputs, i.e.,

$$\| \boldsymbol{\delta} \|_\infty \leq \epsilon, \quad s.t., \quad \mathbb{P}\big(\text{argmax}_j(\mathcal{D}_\psi(\boldsymbol{x}')_j) \neq \text{argmax}_j(\mathcal{D}_\psi(\boldsymbol{x})_j)\big) > f_r, \tag{1}$$

where, $f_r$ is the fooling ratio, $y$ is the ground-truth label for the example $\boldsymbol{x}$ and the predictions on clean images $\boldsymbol{x}$ are given by, $y = \text{argmax}_j(\mathcal{D}_\psi(\boldsymbol{x})_j)$. Note that we do not necessarily require the ground-truth labels of source domain images to craft a successful attack. In the case of adversarial attacks based on a traditional GAN framework, the following objective is maximized for the generator to achieve the maximal fooling rate:

$$\boldsymbol{\theta}^* \leftarrow \underset{\boldsymbol{\theta}}{\text{argmax}} \ \text{CROSSENTROPY}(\mathcal{D}_\psi(\boldsymbol{x}'), \mathbb{1}_y), \tag{2}$$

where $\mathbb{1}_y$ is the one-hot encoded label vector for an input example $\boldsymbol{x}$. The above objective seeks to maximize the discriminator error on the perturbed images that are output from the generator network.

We argue that the objective given by Eq. 2 does not directly enforce transferability for the generated perturbations $\boldsymbol{\delta}$. This is primarily due to the reason that the discriminator's response for clean examples is totally ignored in the conventional generative attacks. Here, inspired by the generative adversarial network in [21], we propose a relativistic adversarial perturbation (RAP) generation approach that explicitly takes in to account the discriminator's predictions on clean images. Alongside reducing the classifier's confidence on perturbed images, the attack algorithm also forces the discriminator to maintain a high confidence scores for the clean samples. The proposed relativistic objective is given by:

$$\boldsymbol{\theta}^* \leftarrow \underset{\boldsymbol{\theta}}{\text{argmax}} \ \text{CROSSENTROPY}(\mathcal{D}_\psi(\boldsymbol{x}') - \mathcal{D}_\psi(\boldsymbol{x}), \mathbb{1}_y). \tag{3}$$

The cross entropy loss would be higher when the perturbed image is scored significantly lower than the clean image response for the ground-truth class i.e., $\mathcal{D}_\psi(\boldsymbol{x}')_y \ll \mathcal{D}_\psi(\boldsymbol{x})_y$. The discriminator basically seeks to increase the 'fooling gap' $(\mathcal{D}_\psi(\boldsymbol{x}')_y - \mathcal{D}_\psi(\boldsymbol{x})_y)$ between the true and perturbed samples. Through such relative discrimination, we not only report better transferability rates across networks trained on the same domain, but most importantly show excellent cross-domain transfer rates for the instance-agnostic perturbations. We attribute this behaviour to the fact that once a perturbation pattern is optimized using the proposed loss on a source distribution (e.g., paintings, cartoon images), the generator learns a "contrastive" signal that is agnostic to the underlying distribution. As a result, when the same perturbation pattern is applied to networks trained on totally different domain (e.g., natural images), it still achieves the state-of-the-art attack transferability rates. Table 2 shows the gain in transferability when using relativistic cross-entropy (Eq. 3) in comparison to simple cross-entropy loss (Eq. 2).

For an untargeted attack, the above mentioned objective in Eq. 2 and 3 suffices, however, for a targeted adversarial attack, the prediction for the perturbed image must match a given target class $y'$ i.e., $\text{argmax}_j(\mathcal{D}_\psi(\boldsymbol{x}')_j) = y' \neq y$. For such a case, we employ the following loss function:

$$\boldsymbol{\theta}^* \leftarrow \underset{\boldsymbol{\theta}}{\text{argmin}} \ \text{CROSSENTROPY}(\mathcal{D}_\psi(\boldsymbol{x}'), \mathbb{1}_{y'}) + \text{CROSSENTROPY}(\mathcal{D}_\psi(\boldsymbol{x}), \mathbb{1}_y). \tag{4}$$

The overall training scheme for the generative network is given in Algorithm 1.

---

**Algorithm 1** Generator Training for Relativistic Adversarial Perturbations

---

1: A pretrained classifier $\mathcal{D}_\psi$, arbitrary training data distribution $\mathcal{X}$, perturbation budget $\epsilon$, loss criteria $\mathcal{L}$.
2: Randomly initialize generator network $\mathcal{G}_\theta$
3: **repeat**
4:     Sample mini-batch of data from the training set.
5:     Use the current state of the generator, $\mathcal{G}_\theta$, to generate unbounded adversaries.
6:     Project adversaries, $\mathcal{G}_\theta(x)$, within a valid perturbation budget to obtain $x'$ such that $\|x' - x\|_\infty \leq \epsilon$.
7:     Forward pass $x'$ to $\mathcal{D}_\psi$ and compute loss given in Eq. (3)/Eq. (4) for targeted/untargeted attack.
8:     Backward pass and update the generator, $\mathcal{G}_\theta$, parameters to maximize the loss.
9: **until** model convergence.

---

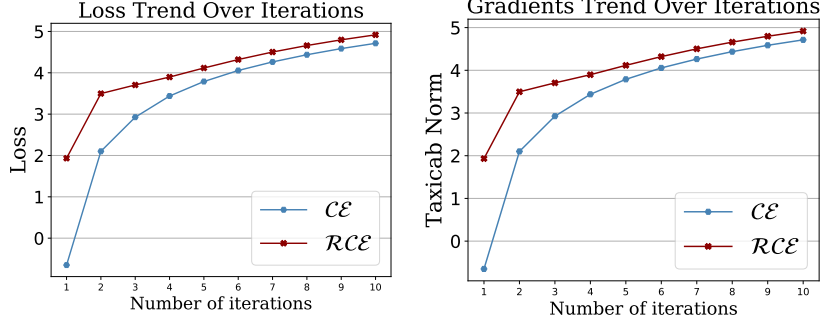

Figure 3: Loss and gradients trend for $\mathcal{CE}$ and $\mathcal{RCE}$ loss functions. Results are reported with VGG16 network on 100 random images for MI-FGSM attack. Trends are shown in log scale.

# 4   Gradient Perspective of Relativistic Cross-Entropy

Adversarial perturbations are crafted via loss function gradients. An effective loss function helps in the generation of perturbations by back-propagating *stronger* gradients. Below, we show that Relativistic Cross-Entropy ($\mathcal{RCE}$) ensures this requisite and thus leads to better performance than regular Cross-Entropy ($\mathcal{CE}$) loss.

Suppose, the logit-space outputs from the discriminator (pretrained classifier) corresponding to a clean image ($x$) and a perturbed image ($x'$) are denoted by $a$ and $a'$, respectively. Then, $\mathcal{CE}(a', y) = -\log\left(e^{a'_y} / \sum_k e^{a'_k}\right)$ is the cross-entropy loss for a perturbed input $x'$. For clarity, we define $p'_y = e^{a'_y} / \sum_k e^{a'_k}$. The derivative of $p'_y$ w.r.t $a'_i$ is $\partial p'_y / \partial a'_i = p'_y(\llbracket i{=}y \rrbracket - p'_i)$. Using chain rule, the derivative of cross-entropy loss is given by:

$$\frac{\partial \mathcal{CE}}{\partial a'_i} = p'_i - \llbracket i{=}y \rrbracket. \tag{5}$$

For the relativistic loss formulated as $\mathcal{RCE}(a', a, y) = -\log\left(e^{a'_y - a_y} / \sum_k e^{a'_k - a_k}\right)$, we define $r_y = \left(e^{a'_y - a_y} / \sum_k e^{a'_k - a_k}\right)$. The derivative of $r_y$ w.r.t $a'_i$ is $\partial r_y / \partial a'_i = r_i(\llbracket i{=}y \rrbracket - r_y)$. From chain rule, $\mathcal{RCE}$ derivative w.r.t to $a'_i$ is given by:

$$\frac{\partial \mathcal{RCE}}{\partial a'_i} = r_i - \llbracket i{=}y \rrbracket. \tag{6}$$

In light of above relation, $\mathcal{RCE}$ has three important properties:

1. Comparing (Eq.5) with (Eq.6) shows that $\mathcal{RCE}$ gradient is a function of '*difference*' $(a'_y - a_y)$ as opposed to only scores $a'_y$ in $\mathcal{CE}$ loss. Thus, it measures the relative change in prediction as an explicit objective during optimization.

2. $\mathcal{RCE}$ loss back-propagates larger gradients compared to $\mathcal{CE}$, resulting in efficient training and stronger adversaries (see Figure 3 for empirical evidence). **Sketch Proof:** We can factorize the denominator in (Eq. 6) as follows: $\partial \mathcal{RCE}/\partial a'_i = \left(e^{a'_y - a_y} / \left(e^{a'_y - a_y} + \sum_{k \neq y} e^{a'_k - a_k}\right)\right) - \llbracket i{=}y \rrbracket$. Consider the fact that maximization of $\mathcal{RCE}$ is only possible when $e^{(a'_y - a_y)}$ decreases

and $\sum_{k \neq y} e^{(a'_k - a_k)}$ increases. Generally, $a_y \gg a_{k \neq y}$ for the score generated by a pre-trained model and $a'_y \ll a'_{k \neq y}$ (here $k$ denotes an incorrectly predicted class). Thus, $\partial \mathcal{RCE}/\partial a'_i > \partial \mathcal{CE}/\partial a'_i$ since $e^{(a'_y - a_y)} < e^{(a'_y)}$ and $\sum_{k \neq y} e^{(a'_k - a_k)} > \sum_{k \neq y} e^{(a'_k)}$. In simple words, the gradient strength of $\mathcal{RCE}$ is higher than $\mathcal{CE}$.

3. In case $x$ is misclassified by $\mathcal{F}(\cdot)$, the gradient strength of $\mathcal{RCE}$ is still higher than $\mathcal{CE}$ (here noise update with the $\mathcal{CE}$ loss will be weaker since adversary's goal is already achieved i.e., $x$ is misclassified).

| Loss | VGG-16 | VGG-19 | Squeeze-v1.1 | Dense-121 |
|------|--------|--------|--------------|-----------|
| Cross Entropy (CE) | 79.21 | 78.96 | 69.32 | 66.45 |
| Relativistic CE | **86.95** | **85.88** | **77.81** | **75.21** |

Table 2: Effect of Relativistic loss on transferability in terms of fooling rate (%) on ImageNet val-set. Generator is trained against ResNet-152 on Paintings dataset.

# 5 Experiments

## 5.1 Rules of the Game

We report results using following three different attack settings in our experiments: **(a) White-box.** Attacker has access to the original model (both architecture and parameters) and the training data distribution. **(b) Black-box.** Attacker has access to a pretrained model on the same distribution but without any knowledge of the target architecture and target data distribution. **(c) Cross-domain Black-box.** Attacker has neither access to (any) pretrained model, nor to its label space and its training data distribution. It then has to seek a transferable adversarial function that is learned from a model pretrained on a possibly different distribution than the original. Hence, this setting is relatively far more challenging than the plain black-box setting.

| Perturbation | Attack | VGG-19 | | ResNet-50 | | Dense-121 | |
|--------------|--------|--------------------|-------------------|--------------------|-------------------|--------------------|-------------------|
| | | Fool Rate ($\uparrow$) | Top-1 ($\downarrow$) | Fool Rate ($\uparrow$) | Top-1 ($\downarrow$) | Fool Rate ($\uparrow$) | Top-1 ($\downarrow$) |
| $l_\infty \leq 10$ | Gaussian Noise | 23.59 | 64.65 | 18.06 | 70.74 | 17.05 | 70.30 |
| | Ours-Paintings | 47.12 | 46.68 | 31.52 | 60.77 | 29.00 | 62.0 |
| | Ours-Comics | **48.47** | **45.78** | **33.69** | **59.26** | **31.81** | **60.40** |
| | Ours-ChestX | 40.81 | 50.11 | 22.00 | 67.72 | 20.53 | 67.63 |
| $l_\infty \leq 16$ | Gaussian Noise | 33.80 | 57.92 | 25.76 | 66.07 | 23.30 | 66.70 |
| | Ours-Paintings | 66.52 | 30.21 | 47.51 | 47.62 | 44.50 | 49.76 |
| | Ours-Comics | **67.75** | **29.25** | **51.78** | **43.91** | **50.37** | **45.17** |
| | Ours-ChestX | 62.14 | 33.95 | 34.49 | 58.6 | 31.81 | 59.75 |
| $l_\infty \leq 32$ | Gaussian Noise | 61.07 | 35.48 | 47.21 | 48.40 | 39.90 | 54.37 |
| | Ours-Paintings | 87.08 | 11.96 | 69.05 | 28.77 | 63.78 | 33.46 |
| | Ours-Comics | 87.90 | 11.17 | **71.91** | **26.12** | **71.85** | **26.18** |
| | Ours-ChestX | **88.12** | **10.92** | 62.17 | 34.85 | 59.49 | 36.98 |

Table 3: **Cross-Domain Black-box**: Untargeted attack success (%) in terms of fooling rate on ImageNet val-set. Adversarial generators are trained against ChexNet on Paintings, Comics and ChestX datasets. Perturbation budget, $l_\infty \leq 10/16/32$, is chosen as per the standard practice. Even without the knowledge of targeted model, its label space and its training data distribution, the transferability rate is much higher than the Gaussian noise.

## 5.2 Experimental Settings

**Generator Architecture.** We chose ResNet architecture introduced in [22] as the generator network $\mathcal{G}_{\theta}$; it consists of downsampling, residual and upsampling blocks. For training, we used Adam optimizer [23] with a learning rate of 1e-4 and values of exponential decay rate for first and second moments set to 0.5 and 0.999, respectively. Generators are learned against the four pretrained ImageNet models including VGG-16, VGG-19 [24], Inception (Inc-v3) [25], ResNet-152 [26] and ChexNet (which is a Dense-121 [27] network trained to diagnose pneumonia) [28].

**Datasets.** We consider the following datasets for generator training namely Paintings [29], Comics [30], ImageNet and a subset of ChestX-ray (ChestX) [28]. There are approximately 80k samples in Paintings, 50k in Comics, 1.2 million in ImageNet training set and 10k in ChestX.

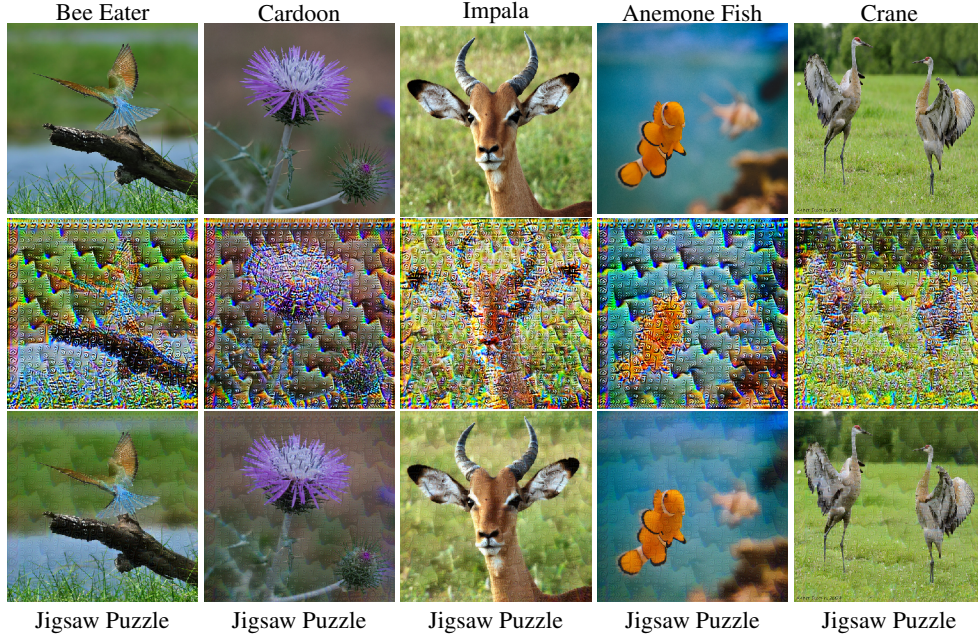

Figure 4: Untargeted adversaries produced by generator trained against Inception-v3 on Paintings dataset. 1st row shows original images while 2nd row shows unrestricted outputs of adversarial generator and 3rd row are adversaries after valid projection. Perturbation budget is set to $l_{\infty} \leq 10$.

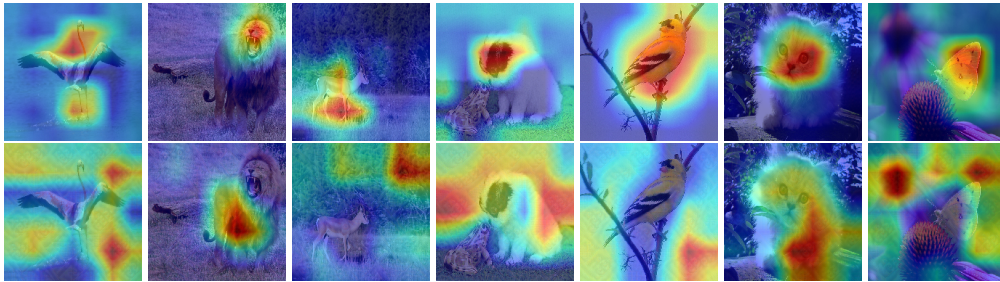

Figure 5: Illustration of attention shift. We use [32] to visualize attention maps of clean (1st row) and adversarial (2nd row) images. Adversarial images are obtained by training generator against VGG-16 on Paintings dataset.

**Inference:** Inference is performed on ImageNet validation set (val-set) (50k samples), a subset (5k samples) of ImageNet proposed by [11] and ImageNet-NeurIPS [31] (1k samples) dataset.

**Evaluation Metrics:** We use the fooling rate (percentage of input samples for which predicted label is flipped after adding adversarial perturbations), top-1 accuracy and % increase in error rate (the difference between error rate of adversarial and clean images) to evaluate our proposed approach.

### 5.2.1 Results

Table 3 shows the cross-domain black-box setting results, where attacker have no access to model architecture, parameters, its training distribution or label space. Note that ChestX [28] does not have much texture, an important feature to deceive ImageNet models [33], yet the transferability rate of perturbations learned against ChexNet is much better than the Gaussian noise.

Tables 4 and 5 show the comparison of our method against different universal methods on both naturally and adversarially trained models [34] (Inc-v3, Inc-v4 and IncRes-v2). Our attack success rate is much higher both in white-box and black-box settings. Notably, for the case of adversarially trained models, Gaussian smoothing on top of our approach leads to significant increase in transferability. We provide further comparison with GAP [16] in the supplementary material. Figures 4 and 5 show the model's output and attention shift on example adversaries.

| Model | Attack | VGG-16 | VGG-19 | ResNet-152 |
|---|---|---|---|---|
| VGG-16 | FFF | 47.10* | 41.98 | 27.82 |
| | AAA | 71.59* | 65.64 | 45.33 |
| | UAP | 78.30* | 73.10 | **63.40** |
| | Ours-Paintings | 99.58* | 98.97 | 47.90 |
| | Ours-Comics | **99.83*** | **99.56** | 58.18 |
| | Ours-ImageNet | 99.75* | 99.44 | 52.64 |
| VGG-19 | FFF | 38.19 | 43.60* | 26.34 |
| | AAA | 69.45 | 72.84* | 51.74 |
| | UAP | 73.50 | 77.80* | **58.00** |
| | Ours-Paintings | 98.90 | 99.61* | 40.98 |
| | Ours-Comics | **99.29** | 99.76* | 42.61 |
| | Ours-ImageNet | 99.19 | **99.80*** | 53.02 |
| ResNet-152 | FFF | 19.23 | 17.15 | 29.78* |
| | AAA | 47.21 | 48.78 | 60.72* |
| | UAP | 47.00 | 45.5 | 84.0* |
| | Ours-Paintings | 86.95 | 85.88 | 98.03* |
| | Ours-Comics | 88.94 | 88.84 | 94.18* |
| | Ours-ImageNet | **95.40** | **93.26** | **99.02*** |

Table 4: **White & Black-box Setting:** Fool rate (%) of untargeted attack on ImageNet val-set. Perturbation budget is $l_\infty \leq 10$. * indicates white-box attack. Our attack's transferability from ResNet-152 to VGG-16/19 is even higher than other white-box attacks.

| Model | Attack | Inc-v3$_{ens3}$ | Inc-v3$_{ens4}$ | IncRes-v2$_{ens}$ |
|---|---|---|---|---|
| Inc-v3 | UAP | 1.00/7.82 | 1.80/5.60 | 1.88/5.60 |
| | GAP | 5.48/33.3 | 4.14/29.4 | 3.76/22.5 |
| | RHP | 32.5/60.8 | 31.6/58.7 | 24.6/57.0 |
| Inc-v4 | UAP | 2.08/7.68 | 1.94/6.92 | 2.34/6.78 |
| | RHP | 27.5/60.3 | 26.7/62.5 | 21.2/58.5 |
| IncRes-v2 | UAP | 1.88/8.28 | 1.74/7.22 | 1.96/8.18 |
| | RHP | 29.7/62.3 | 29.8/63.3 | 26.8/62.8 |
| Ours-Paintings | | 33.92/72.46 | 38.94/71.4 | 33.24/69.66 |
| Ours-gs-Paintings | | **47.78/73.06** | **48.18/72.68** | **42.86/73.3** |
| Ours-Comics | | 21.06/67.5 | 24.1/68.72 | 12.82/54.72 |
| Ours-gs-Comics | | 34.52/70.3 | 56.54/69.9 | 23.58/68.02 |
| Ours-ImageNet | | 28.34/71.3 | 29.9/66.72 | 19.84/60.88 |
| Ours-gs-ImageNet | | 41.06/71.96 | 42.68/71.58 | 37.4/72.86 |

Table 5: **Black-box Setting:** Transferability comparison in terms of % increase in error rate after attack. Results are reported on subset of ImageNet (5k) with perturbation budget of $l_\infty \leq 16/32$. Our generators are trained against naturally trained Inc-v3 only. 'gs' represents Gaussian smoothing applied to generator output before projection that enhances our attack strength.

### 5.2.2 Comparison with State-of-the-Art

Finally, we compare our method with recently proposed instance-specific attack method [10] that exhibits high transferability to adversarially trained models. For the very first time in literature, we showed that a universal function like ours can attain much higher transferability rate, outperforming the state-of-the-art instance-specific translation invariant method [10] by a large average absolute gain of 46.6% and 86.5% (in fooling rates) on both naturally and adversarially trained models, respectively, as reported in Table 6. The naturally trained models are Inception-v3 (Inc-v3) [25], Inception-v4 (Inc-v4), Inception Resnet v2 (IncRes-v2) [35] and Resnet v2-152 (Res-152) [36]). The adversarially trained models are from [34].

| | Attack | Naturally Trained | | | | Adversarially Trained | | |
|---|---|---|---|---|---|---|---|---|
| | | Inc-v3 | Inc-v4 | IncRes-v2 | Res-152 | Inc-v3$_{ens3}$ | Inc-v3$_{ens4}$ | IncRes-v2$_{ens}$ |
| Inc-v3 | FGSM | 79.6* | 35.9 | 30.6 | 30.2 | 15.6 | 14.7 | 7.0 |
| | TI-FGSM | 75.5* | 37.3 | 32.1 | 34.1 | 28.2 | 28.9 | 22.3 |
| | MI-FGSM | 97.8* | 47.1 | 46.4 | 38.7 | 20.5 | 17.4 | 9.5 |
| | TI-MI-FGSM | 97.9* | 52.4 | 47.9 | 41.1 | 35.8 | 35.1 | 25.8 |
| | DIM | 98.3* | 73.8 | 67.8 | 58.4 | 24.2 | 24.3 | 13.0 |
| | TI-DIM | 98.5* | 75.2 | 69.2 | 59.2 | 46.9 | 47.1 | 37.4 |
| IncRes-v2 | FGSM | 44.3 | 36.1 | 64.3* | 31.9 | 18.0 | 17.2 | 10.2 |
| | TI-FGSM | 49.7 | 41.5 | 63.7* | 40.1 | 34.6 | 34.5 | 27.8 |
| | MI-FGSM | 74.8 | 64.8 | 100.0* | 54.5 | 25.1 | 23.7 | 13.3 |
| | TI-MI-FGSM | 76.1 | 69.5 | 100.0* | 59.6 | 50.7 | 51.7 | 49.3 |
| | DIM | 86.1 | 83.5 | 99.1* | 73.5 | 41.2 | 40.0 | 27.9 |
| | TI-DIM | 86.4 | 85.5 | 98.8* | 76.3 | 61.3 | 60.1 | 59.5 |
| Res-152 | FGSM | 40.1 | 34.0 | 30.3 | 81.3* | 20.2 | 17.7 | 9.9 |
| | TI-FGSM | 46.4 | 39.3 | 33.4 | 78.9* | 34.6 | 34.5 | 27.8 |
| | MI-FGSM | 54.2 | 48.1 | 44.3 | 97.5* | 25.1 | 23.7 | 13.3 |
| | TI-MI-FGSM | 55.6 | 50.9 | 45.1 | 97.4* | 39.9 | 37.7 | 32.8 |
| | DIM | 77.0 | 77.8 | 73.5 | 97.4* | 40.5 | 36.0 | 24.1 |
| | TI-DIM | 77.0 | 73.9 | 73.2 | 97.2* | 60.3 | 58.8 | 42.8 |
| | Ours-Paintings | **100.0*** | 99.7 | **99.8** | **98.9** | 69.3 | 74.6 | 64.8 |
| | Ours-gs-Paintings | 99.9* | 98.5 | 97.6 | 93.6 | **85.2** | **83.9** | **75.9** |
| | Ours-Comics | 99.9* | **99.8** | **99.8** | 98.7 | 39.3 | 46.8 | 23.3 |
| | Ours-gs-Comics | 99.9* | 97.0 | 93.4 | 87.7 | 60.3 | 58.8 | 42.8 |
| | Ours-ImageNet | 99.8* | 99.1 | 97.5 | 98.1 | 55.4 | 60.5 | 36.4 |
| | Ours-gs-ImageNet | 98.9* | 95.4 | 90.5 | 91.8 | 78.6 | 78.4 | 68.9 |

Table 6: **White-box and Black-box:** Transferability comparisons. Success rate on ImageNet-NeurIPS validation set (1k images) is reported by creating adversaries within the perturbation budget of $l_\infty \leq 16$, as per the standard practice [10]. Our generators are learned against naturally trained Inception-v3 only. * indicates white-box attack. 'gs' is Gaussian smoothing applied to the generator output before projection. Smoothing leads to slight decrease in transferability on naturally trained but shows significant increase against adversarially trained models.

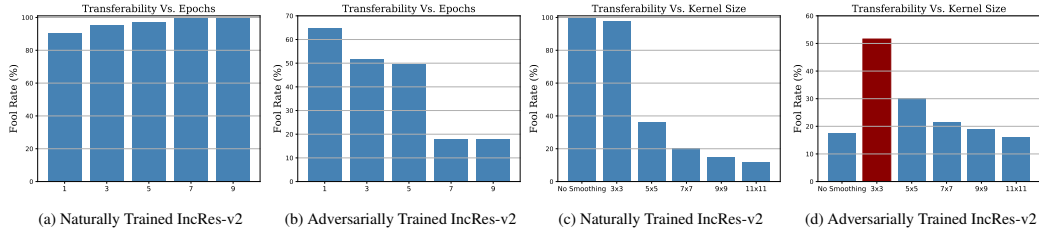

Figure 6: Effect of Gaussian kernel size and number of training epochs is shown on the transferability (in %age fool rate) of adversarial examples. Generator is trained against Inception-v3 on Paintings, while the inference is performed on ImageNet-NeurIPS. Firstly, as number of epochs increases, transferability against naturally trained IncRes-v2 increases while decreases against its adversarially trained version. Secondly, as the size of Gaussian kernel increases, transferability against naturally as well as adversarially trained IncRes-v2 decreases. Applying kernel of size 3 leads to optimal results against adversarially trained model. Perturbation is set to $l_\infty \leq 16$.

## 5.3 Transferability: Naturally Trained vs. Adversarially Trained

Furthermore, we study the impact of training iterations and Gaussian smoothing [10] on the transferability of our generative adversarial examples. We report results using naturally and adversarially trained IncRes-v2 model [35] as other models exhibit similar behaviour. Figure 6 displays the transferability (in %age accuracy) as a function of the number of training epochs (a-b) and various kernel sizes for Gaussian smoothing (c-d).

Firstly, we observe a gradual increase in the transferability of generator against the naturally trained model as the training epochs advance. In contrast the transferability deteriorates against the adversarially trained model. Therefore, when targeting naturally trained models, we train for ten epochs on Paintings, Comics, and ChestX datasets (although we anticipate better performance for higher epochs). When targeting adversarially trained models, we deploy an early stopping criterion to obtain the best trained generator since the performance drops on such models as epochs are increased. This fundamentally shows the reliance of naturally and adversarially trained models on different set of features. Our results clearly demonstrate that the adversarial solution space is shared across different architectures and even across distinct data domains. Since we train our generator against naturally trained models only, therefore it converges to a solution space on which an adversarially trained model has already been trained. As a result, our perturbations gradually become weaker against adversarially trained models as the training progress. A visual demonstration is provided in supplementary material.

Secondly, the application of Gaussian smoothing reveals different results on naturally trained and adversarially trained models. After applying smoothing, adversaries become stronger for adversarially trained models and get weaker for naturally trained models. We achieve optimal results with the kernel size of 3 and $\sigma = 1$ for adversarially trained models and use these settings consistently in our experiments. We apply Gaussian kernel on the unrestricted generator's output, therefore as the kernel size is increased, generator's output becomes very smooth and after projection within valid $l_\infty$ range, adversaries become weaker.

## 6 Conclusion

Adversarial examples have been shown to be transferable across different models trained on the same domain. For the first time in literature, we show that the cross-domain transferable adversaries exists that can fool the target domain networks with high success rates. We propose a novel generative framework that learns to generate strong adversaries using a relativistic discriminator. Surprisingly, our proposed universal adversarial function can beat the instance-specific attack methods that were previously found to be much stronger compared to the universal perturbations. Our generative attack model trained on Chest X-ray and Comics images, can fool VGG-16, ResNet50 and Dense-121 models with a success rate of $\sim 88\%$ and $\sim 72\%$, respectively, without having any knowledge of data distribution or label space.

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
