[Supplementary Material]

# Supplementary: Cross Domain Transferability of Adversarial Perturbations

We further validate the significance of $\mathcal{RCE}$ compared to $\mathcal{CE}$ in terms of three criterion: accuracy, logits difference and transfer to unseen classes (see Figure 1). For the test on unseen classes, we divide ImageNet into two mutually exclusive sets (500 classes each), named IN1 and IN2. VGG16 and ResNet50 are trained on IN1 & IN2 from scratch. We also compare our method with GAP [16] in Sec. 1 to demonstrate superiority of our approach. In Sec. 2, we visually demonstrate the effect of training time and Gaussian kernel size of the generated adversaries. Finally, in Sec. 3, we show adversaries produced by different generators as well as demonstrate attention shift on adversarial examples.

Figure 1: (a) shows Top-5 accuracy of adversaries (*lower is better*), (b) shows normalized $l_2$ difference b/w logits of adversarial and benign examples (*higher is better*) while (c) shows transferability to unseen classes. In each case $\mathcal{RCE}$ performs significantly better than $\mathcal{CE}$. ImageNet val. 50k images are used in (a) and (b) while 25k validation images of IN1 and IN2 are used in (c) and (d).

## 1    Comparison with GAP [16]

| Perturbation | Attack | VGG-16 | | VGG-19 | | Inception-v3 | |
|---|---|---|---|---|---|---|---|
| | | Fool Rate ($\uparrow$) | Top-1 ($\downarrow$) | Fool Rate ($\uparrow$) | Top-1 ($\downarrow$) | Fool Rate ($\uparrow$) | Top-1 ($\downarrow$) |
| $l_\infty \leq 7$ | GAP | 66.9 | 30.0 | 68.4 | 28.8 | 85.3 | 13.7 |
| | Ours-Paintings | 95.31 | 4.29 | 96.84 | 2.94 | 97.95 | 1.86 |
| | Ours-Comics | 99.15 | 0.97 | 98.58 | 1.33 | 98.90 | 1.0 |
| | Ours-ImageNet | 98.57 | 1.32 | 98.71 | 1.24 | 91.03 | 8.4 |
| $l_\infty \leq 10$ | GAP | 80.80 | 17.7 | 84.10 | 14.6 | 98.3 | 1.7 |
| | Ours-Paintings | 99.58 | 0.4 | 99.61 | 0.38 | 99.65 | 0.33 |
| | Ours-Comics | 99.83 | 0.16 | 99.76 | 0.22 | 99.72 | 0.26 |
| | Ours-ImageNet | 99.75 | 0.24 | 99.80 | 0.21 | 99.05 | 0.89 |
| $l_\infty \leq 13$ | GAP | 88.5 | 10.6 | 90.7 | 8.6 | 99.5 | 0.5 |
| | Ours-Paintings | 99.86 | 0.16 | 99.83 | 0.16 | 99.8 | 0.18 |
| | Ours-Comics | 99.88 | 0.12 | 99.86 | 0.13 | 99.83 | 0.17 |
| | Ours-ImageNet | 99.87 | 0.13 | 99.86 | 0.15 | 99.67 | 0.13 |

Table 1: Comparison between GAP [16] and our method. Untargeted attack success rate (%) in terms of fooling rate (*higher is better*) and Top-1 accuracy (*lower is better*) is reported on 50k validation images. Each attack is carried out in a white-box setting.

## 2    Effect of Training Time and Gaussian Kernel Size

Figures 2 and 3 show the evolution of generative adversaries as the number of epochs increases. At initial epochs, adversaries are more smoother and more transferable against adversarially trained models. On the other hand, as training progress, generator converges to a solution with locally strong patterns that are more transferable to naturally trained models.

Figures 4 and 5 show the effect of Gaussian smoothing. As the kernel size increases, transferability of adversaries decreases.

Figure 2: Evolution of adversaries produced by generator as the training progress. Adversaries found at initial training stages e.g., at epoch #1 are highly transferable against adversarially trained models while adversaries found at later training stage e.g., at epoch #10 are highly transferable against naturally trained models. Generator is trained against Inc-v3 on Paintings dataset. First row shows unrestricted adversaries while second row shows adversaries after valid projection ($l_\infty \leq 10$).

Figure 3: Evolution of adversaries produced by generator as the training progress. Adversaries found at initial training stage e.g., at epoch #1 are highly transferable against adversarially trained models while adversaries found at later training stage e.g., at epoch #10 are highly transferable against naturally trained models. Generator is trained against Inc-v3 on Paintings dataset. First row shows unrestricted adversaries while second row shows adversaries after valid projection ($l_\infty \leq 10$).

# 3 Examples

Figure 6 demonstrates the attention shift on generative adversarial examples produced by our method. Figures 7, 8, 9 and 10 show examples of different clean images and their corresponding adversaries produced by different generators.

Figure 4: Evolution of adversaries produced by generator as the size of Gaussian kernel increases. Adversaries start to lose their effect as the kernel size increase. The optimal results against adversarially trained models are found at kernel size of 3. First and second rows show unrestricted adversaries before and after smoothing, while third row shows adversaries after valid projection ($l_\infty \leq 10$).

Figure 5: Evolution of adversaries produced by generator as the size of Gaussian kernel increases. Adversaries start to lose their effect as the kernel size increase. The optimal results against adversarially trained models are found at kernel size of 3. First and second rows show unrestricted adversaries before and after smoothing, while third row shows adversaries after valid projection ($l_\infty \leq 10$).

Figure 6: Illustration of attention shift for ResNet-152. We use [32] to visualize attention maps of clean (1st row) and adversarial (2nd row) images. Adversarial images are obtained by training generator against ResNet-152 on Paintings dataset.

Original Images

Target model: VGG-16, Distribution: Paintings, Fooling rate: 99.58%

Target model: VGG-16, Distribution: Comics, Fooling rate: 99.8%

Target model: VGG-16, Distribution: ImageNet, Fooling rate: 99.7%

Figure 7: Untargeted adversaries produced by generator (before and after projection) trained against VGG-16 on different distributions (Paintings, Comics and ImageNet). Perturbation budget is set to $l_\infty \leq 10$ and fooling rate is reported on ImageNet validation set.

Original Images

Target model: VGG-19, Distribution: Paintings, Fooling rate: 99.6%

Target model: VGG-19, Distribution: Comics, Fooling rate: 99.76%

Target model: VGG-19, Distribution: ImageNet, Fooling rate: 99.8%

Figure 8: Untargeted adversaries produced by generator (before and after projection) trained against VGG-19 on different distributions (Paintings, Comics and ImageNet). Perturbation budget is set to $l_\infty \leq 10$ and fooling rate is reported on ImageNet validation set.

Original Images

Target model: Inc-v3, Distribution: Paintings, Fooling rate: 99.65%

Target model: Inc-v3, Distribution: Comics, Fooling rate: 99.72%

Target model: Inc-v3, Distribution: ImageNet, Fooling rate: 99.04%

Figure 9: Untargeted adversaries produced by generator (before and after projection) trained against Inception-v3 on different distributions (Paintings, Comics and ImageNet). Perturbation budget is set to $l_\infty \leq 10$ and fooling rate is reported on ImageNet validation set.

Original Images

Target model: ResNet-152, Distribution: Paintings, Fooling rate: 98.0%

Target model: ResNet-152, Distribution: Comics, Fooling rate: 94.18%

Target model: ResNet-152, Distribution: ImageNet, Fooling rate: 99.0%

Figure 10: Untargeted adversaries produced by generator (before and after projection) trained against ResNet-152 on different distributions (Paintings, Comics and ImageNet). Perturbation budget is set to $l_\infty \leq 10$ and fooling rate is reported on ImageNet validation set.