[Reviews · NeurIPS 2019]

Reviewer 1



- Clarity: the paper is very well written. The problem is clearly formulated with appropriate distinction with other methods in the abstract and introduction. Compared to the prior works, conceptual differences are well stated. I enjoyed reading it. - Originality and significance: despite the impressive empirical results, the main novelty of this work might be applying the idea of relativistic loss from [20] to adversarial perturbation. I do not underestimate the effort of applying a new technique to a different problem. Then, not only empirical results but also more follow-up analyses should be followed to validate how and why it is applicable. For example, I am curious about what types of cross-domain features are learned across domains. I get a little bit of hint from Figure 3: the background texture becomes more painting styles, but they are not enough. Depending on the combination of different source and target domains, how could the cross-domain features be transferable or not? The predictions on adversarial examples were all Jigsaw Puzzle. Does that indicate that any features in Painting are biased to the class? More broadly, how does style transfer relate to this cross-domain perturbation? Please provide us your insights on bigger and scientific questions.

Reviewer 2



Originality: The authors propose to apply the relativistic discriminator loss to the generation of adversarial examples. The novelty is limited. Significance: The proposed method shows superior performance than several existing method in the field. Clarity: The presentation is clear overall, but there may be several typos in notation which may cause some confusion. For example, D(x, x′) in line 107 is misleading, as D takes in a single input alone. Quality: The manuscript lacks enough theoretical arguments / explanations why the relativistic discriminator loss is better than the traditional cross-entropy loss in generating adversarial examples. On the other hand, sufficient experimental results are provided for validating this claim. Pros: The paper provides sufficient experimental results for validating the claim that the proposed relativistic discriminator loss helps the training of a transferable generator of adversarial examples. It also provides a very concrete introduction and related work in terms of different attacking settings (or threat models). Cons: A main problem with the paper may be the lack of novelty. The proposed relativistic discriminator seems to be only a small modification of an existing RGANs. While the authors successfully applied this model to generation of transferable adversarial examples, they fail to explain the fundamental reason why this loss has a benefit.

Reviewer 3



This submission proposes to use a GAN to generate adversarial examples. The key difference from prior work is that the discriminator uses a relativistic loss (and perturbations are projected to satisfy l_infinity norm <= 10, rather than being unbounded). This relativistic loss enables the generator to create successful cross-domain adversarial examples; in other words, it can generate adversarial examples for an image distribution (e.g., ImageNet) that it was not trained on. The use of a relativistic loss in training GANs is not novel, as the authors acknowledge, but this is the first time it has been applied to generating adversarial examples, and the result is quite impressive. It leads to state-of-the-art adversarial attacks on both naturally-trained and adversarially-trained ImageNet models, in *both* white-box and black-box settings. Originality and Significance: This submission is low in originality, but high in significance. Although the contribution has relatively low technical novelty, the fact that it leads to state-of-the-art adversarial attacks is highly significant. The community should be aware of this, so that it can take these attacks into consideration when generating defenses. In terms of related work, the submission did a good job of acknowledging existing work on adversarial attacks that are image-agnostic or produced by generative models. One recent work (in the latter category) that should be included is [1]. When comparing with prior methods (those in Table 1), why not also compare to other methods that use GANs, for instance that work or [19]? I would also appreciate a more detailed comparison of how the proposed GAN framework differs from those two works. Clarity: I found the submission to be clearly written and well-organized. Quality: I'm curious about the use of "instance-agnostic" to describe adversarial examples produced by GANs. Is this common terminology? There is still a separate adversarial perturbation computed per image, rather than a single perturbation that is added to all images (as in UAP). It's true that only a single forward pass is necessary, instead of a backward pass as well, or multiple forward and backward passes. But it doesn't seem truly instance-agnostic to me. [1] Song et al. Constructing Unrestricted Adversarial Examples with Generative Models. NeurIPS 2018. https://arxiv.org/abs/1805.07894

[Author Response · NeurIPS 2019]

We thank the reviewers for the positive feedback: new state-of-the-art results
(R1,2&3), first to explore cross-domain transferability (R1), high significance
to the community (R3), very well written and clear presentation (R1,2&3).
**Code:** Code will be made public. Fig.( 1, 2, 3) best viewed in zoom.

Figure 1: Loss and gradients trend for CE and RCE loss. Results are reported with VGG16 network on 100 random images for MIFGSM attack.

**R1,2&3: Significance of Relativistic Cross-Entropy (RCE):** Adversarial per-
turbations are crafted via loss function gradients. An effective loss helps in
adversary generation by back-propagating *stronger* gradients. Below, we show
that $\mathcal{RCE}$ ensures this requisite and thus leads to better performance than $\mathcal{CE}$.
**Notation:** classifier $\mathcal{F}$, clean sample $\boldsymbol{x}$, adversarial example $\boldsymbol{x}'$, output scores $a = \mathcal{F}(\boldsymbol{x})$, $a' = \mathcal{F}(\boldsymbol{x}')$.
**Gradient Perspective:** Let $\mathcal{CE}(a', y) = -\log\left(e^{a'_y}/\sum_k e^{a'_k}\right)$ be the CE loss for input $\boldsymbol{x}'$. For clarity, we define
$p'_y = e^{a'_y}/\sum_k e^{a'_k}$. The derivative of $p'_y$ w.r.t $a'_i$ is $\partial p'_y/\partial a'_i = p'_y(\llbracket i=y \rrbracket - p'_i)$. From chain rule, $\partial \mathcal{CE}/\partial a'_i = p'_i - \llbracket i=y \rrbracket$
(Eq. 1). For relativistic loss, $\mathcal{RCE}(a', a, y) = -\log\left(e^{a'_y - a_y}/\sum_k e^{a'_k - a_k}\right)$, we define $r_y = \left(e^{a'_y - a_y}/\sum_k e^{a'_k - a_k}\right)$. The
derivative of $r_y$ w.r.t $a'_i$ is $\partial r_y/\partial a'_i = r_i(\llbracket i=y \rrbracket - r_y)$. From chain rule, $\partial \mathcal{RCE}/\partial a'_i = r_i - \llbracket i=y \rrbracket$ (Eq. 2).
In light of above relations, $\mathcal{RCE}$ has three important properties: **(a)** Comparing (Eq. 2) with (Eq. 1) shows that $\mathcal{RCE}$ gra-
dient is a function of '*difference*' $(a'_y - a_y)$ as opposed to only scores $a'_y$ in $\mathcal{CE}$ loss. Thus it measures the relative change
in prediction as an explicit objective during optimization. **(b)** $\mathcal{RCE}$ loss back-propagates larger gradients compared to
$\mathcal{CE}$, resulting in efficient training and stronger adversaries (see Fig. 1 for empirical evidence). **Sketch Proof:** We can
factorize the denominator in (Eq. 2) as follows: $\partial \mathcal{RCE}/\partial a'_i = \left(e^{a'_y - a_y}/\left(e^{a'_y - a_y} + \sum_{k \neq y} e^{a'_k - a_k}\right)\right) - \llbracket i=y \rrbracket$. Consider
the fact that maximization of $\mathcal{RCE}$ is only possible when $e^{(a'_y - a_y)}$ decreases and $\sum_{k \neq y} e^{(a'_k - a_k)}$ increases. Generally,
$a_y \ggg a_{k \neq y}$ for the score generated by a pre-trained model and $a'_y \lll a'_{k \neq y}$. Thus, $\partial \mathcal{RCE}/\partial a'_i > \partial \mathcal{CE}/\partial a'_i$ since
$e^{(a'_y - a_y)} < e^{(a'_y)}$ and $\sum_{k \neq y} e^{(a'_k - a_k)} > \sum_{k \neq y} e^{(a'_k)}$. In simple words, the gradient strength of $\mathcal{RCE}$ is higher than $\mathcal{CE}$.
**(c)** In case $\boldsymbol{x}$ is misclassified by $\mathcal{F}(\cdot)$, the gradient strength of $\mathcal{RCE}$ is still higher than $\mathcal{CE}$ (here noise update with the
$\mathcal{CE}$ loss will be weaker since adversary's goal is already achieved i.e., $\boldsymbol{x}$ is misclassified). We will add it in final version.
**Evaluation:** We further validate (see Fig. 2) the significance of $\mathcal{RCE}$ compared to $\mathcal{CE}$ in terms of three criterion
(accuracy, logits difference and transfer to unseen classes). For the test on unseen classes, we divide ImageNet into two
mutually exclusive sets (500 classes each), named IN1 and IN2. VGG16 is trained on IN1 & IN2 from scratch.

**R1: 1) RCE Justification:** See R1,2&3 above. **2)**
**Relation with Style-Transfer:** We visualize the
intermediate feature space of cross-domain per-
turbed images and compare it with original and
stylized images (Fig. 3). We note that the feature
space of perturbed images is fairly shifted from
the original and stylized images. This shows that
although some of the generated patterns resemble
"style" of a specific domain (e.g., in Fig. 3 main
paper), the overall behaviour of our proposed ap-

Figure 2: (a) shows Top-5 accuracy of adversaries (lower is better), (b) shows normalized $l_2$ difference b/w logits of adversarial and benign examples (higher is better) while (c) shows transferability to unseen classes. In each case $\mathcal{RCE}$ perform significantly better than $\mathcal{CE}$.

proach is distinct from style transfer. This is potentially due to the existence of "non-robust features" defined as 'features
that are highly predictive but brittle and incomprehensible to humans' [A1]. Since, our generated perturbations are
bounded (as opposed to unbounded style transfer), the attacker is likely to focus on the non-robust features. We will add
further qualitative examples on other domains in final version (Fig. 6 in supp. material). **3) Notations:** Will update in
the final draft. **4) On Adversarial Training Defenses:** Our main draft already includes evaluations with adversarial
training (Tab.5&6 in paper). **5) On the Existence of Universal Adversarial Function (UAF):** Earlier works [A2,A3]
show that universal adversarial perturbations exist due to overlap in decision space of different classification models.
Our work empirically shows that the same holds true even across different domains. This possibly happens due to the
overlap between latent low-dimensional manifolds across different domains.

**R2: Theoretical Result:** See R1,2&3 earlier. **Typo:** We thank R2 & fix it.

**R3: 1) Use of Instance-Agnostic:** We used this term to differentiate the
one-time training feature of our attack as opposed to instance-specific
attacks. However, we acknowledge R3's point and will replace this term
with *domain-agnostic* for clarity. **2) Comparison with [1,19]:** [1] trains
conditional generators to learn original data manifold and searches the latent

Figure 3: t-SNE visualization for features of 100 images and their corresponding stylized and perturbed versions. VGG16 is used to extract features.

space conditioned on the human recognizable target class that is mis-classified by a target classifier. Different to [1],
our approach learns to add adversarial noise to the original samples. [19] produces adversarial images by employing a
separate discriminator alongside classifier. Different to [19], we train a generator to first produce unbounded adversaries
and then project them to nearby original images. We thank R3 and will add further discussion in final version.

**[A1]** Ilyas, Andrew, et al. "Adversarial examples are not bugs, they are features." arXiv (2019). **[A2]** Tramèr, Florian, et al. "The space of transferable adversarial examples." arXiv (2017). **[A3]** Dezfooli, Seyed, et al. "Analysis of universal adversarial perturbations." arXiv (2017).

[Meta-Review · NeurIPS 2019]

The demonstration how transferable perturbations are across domains adds to the intriguing properties we know about neural networks. Employed to perform adversarial attacks the approach presented achieves state of the art results. Unfortunately, given the scores, the paper is not eligible for an oral presentation.